# Biobanking in Molecular Biomarker Research for the Early Detection of Cancer

**DOI:** 10.3390/cancers12040776

**Published:** 2020-03-25

**Authors:** Kim Lommen, Selena Odeh, Chiel C. de Theije, Kim M. Smits

**Affiliations:** 1Department of Pathology, GROW-School for Oncology and Developmental Biology, Maastricht University Medical Center, 6229 HX Maastricht, The Netherlands; k.lommen@maastrichtuniversity.nl (K.L.); s.odeh@maastrichtuniversity.nl (S.O.); 2Central Biobank Maastricht UMC, 6229 ER Maastricht, The Netherlands; chiel.detheije@biobank.nl

**Keywords:** biobank, liquid biopsy, biomarkers, cancer, diagnosis

## Abstract

Although population-wide screening programs for several cancer types have been implemented in multiple countries, screening procedures are invasive, time-consuming and often perceived as a burden for patients. Molecular biomarkers measurable in non-invasively collected samples (liquid biopsies) could facilitate screening, as they could have incremental value on early diagnosis of cancer, but could also predict prognosis or monitor treatment response. Although the shift towards biomarkers from liquid biopsies for early cancer detection was initiated some time ago, there are many challenges that hamper the development of such biomarkers. One of these challenges is large-scale validation that requires large prospectively collected biobanks with liquid biopsies. Establishing those biobanks involves several considerations, such as standardization of sample collection, processing and storage within and between biobanks. In this perspective, we will elaborate on several issues that need to be contemplated in biobanking, both in general and for certain specimen types specifically, to be able to facilitate biomarker validation for early detection of cancer.

## 1. Introduction

Over the past decades, early diagnosis of cancer has become a main focus in research, and population-wide screening programs have been implemented for several cancer types (such as breast, colorectal and cervical cancer) in multiple countries [1]. However, current screening procedures are often invasive or perceived as unpleasant. In addition, they can lead to false positives and might pressure health care, because of time-consuming and costly screening methods. Because of this, many researchers have been aiming to improve cancer screening by focusing on measuring molecular biomarkers in liquid biopsies; non-invasively collected samples, such as stool, blood, sputum, urine or other bodily fluids that are thought to represent the molecular composition of the tumor [2,3]. However, for the clinical translation of biomarkers, large prospectively collected biobanks with corresponding patient data are necessary for biomarker validation after initial publication. Large-scale, independent validation will make sure the biomarker performance can be generalized across populations, which is essential to be able to be translated into clinical practice [4].

Although non-invasive biomarkers for early detection of cancer have become a popular research subject over the past decades, there have been many underlying challenges that hamper their translation into clinical practice. Research indicates that less than 1% of all published biomarkers are eventually implemented in clinical care [5,6]. In agreement with prior publications [4,7], we previously described various problems hindering clinical translation of biomarkers, including a lack of available and appropriate samples, lack of standardized research methodology and a lack of validation [8].

Historically, biobanks were merely gatherings of biological samples supporting specific research projects. Nowadays, biobanks are being established to be ongoing infrastructures with large sophisticated collections of biological samples, complemented by extensive and well-annotated clinical and pathological patient data, sometimes even including medical imaging and pathological histology [9,10]. Depending on the research question, either general or specialized biobanking may be appropriate. General biobanks are often collected population-wide, and are therefore suitable for broad research questions. For specific or rare diseases, a more specialized biobanking approach is needed to ensure that the samples suit the research question and that sufficient samples are available to ensure statistical power.

As modern biobanks use increasingly advanced technology and automated sample processing, and are often not exclusively established to answer specific research questions, large-scale analysis of these samples can be performed for several purposes, making these biobanks more universally applicable. However, samples from different biobanks cannot always be used and interpreted interchangeably, due to different national governmental guidelines regarding patient and data protection [11], but also due to technical differences between biobanks [12], among other things. Various methods for collecting, processing and storing samples, as well as corresponding data, result in heterogeneity between biobanks [12], which can make it more difficult to compare research results from samples originating from existing biobanks.

In this perspective, we will elaborate on several issues that need to be considered when establishing a new biobank, as well as when using an existing biobank, both in general and specific for certain specimen types, to be able to develop and validate biomarkers for early detection of cancer.

## 2. Establishing a Novel Biobank for Molecular Biomarker Research Questions

The commitment of prolonged storage of formalin-fixed paraffin embedded (FFPE) tissue from cancer patients for diagnostic and clinical purposes facilitates researchers in relatively easily obtaining tissue samples, and ample material is available even though collecting the corresponding clinical data may be challenging. Although many liquid biopsies are collected during routine clinical care as well, they are not commonly stored for clinical and future research purposes [13]. Biobanking (specific parts of) these samples could facilitate large-scale validation and clinical translation of liquid biopsy biomarkers for early detection of cancer or other research questions. However, developing biobanks with routinely collected liquid biopsies would require huge efforts, both financially and in workload, and is therefore probably only feasible for specific patient groups and specific research questions. Although implementing biobanking activities in clinical workflows is logistically challenging, several biobanks have described their clinical workflow in illustrative diagrams which could serve as guidelines to others [9,14,15].

### 2.1. Collection, Processing and Storage of Liquid Biopsies for Biobanking

Due to the composition of liquid biopsies, validating candidate biomarkers in these samples has proven to be challenging; some of these challenges need to be considered upfront, when designing the biobank protocol. Depending on sample and biomarker type, both sample properties and protocol components could hinder optimal biobanking and future analyses [16,17,18]. Partially, this could be overcome by carefully selecting stabilization, pretreatment and processing protocols tailored to the future purpose of the samples, and translating this into an optimal logistical process for each biobank (Figure 1). Moreover, it is important to optimize fast processing, short-term storage and stabilization in a way that is logistically convenient for the personnel involved, without harming the sample integrity before long-term storage. To ensure both the quantity (e.g., DNA yield) and the quality (e.g., intact DNA) of the sample, pretreatments of the original sample like centrifugation, or the addition of stabilizing agents to inhibit degradation of the sample, have to be considered. For molecular analysis of DNA and RNA biomarkers for example, it is important that a sufficient amount of DNA or RNA can be yielded from the samples. Therefore, DNases and RNases should preferably be eliminated from the samples to avoid degradation (Figure 1). Addition of, e.g., EDTA for preserving DNA and, e.g., RNAlater for preserving RNA in the sample, should be considered, as these preservatives inhibit DNase and RNase, respectively [19,20], and could thereby facilitate higher DNA or RNA yields from the sample. For molecular biomarkers, PCR-based techniques are commonly used to assess biomarker status. Here, PCR inhibitors (organic or inorganic, soluble or dissolved substances) can disrupt the PCR process at any step, affecting the amplification efficiency and thus resulting in a suboptimal technical assay. Removing PCR inhibitors from the liquid biopsy samples will yield more reliable and reproducible results, but also add another processing step [21,22]. In addition, the DNase/RNase inhibitor of choice, in the chosen concentration, should not act as a PCR inhibitor [22].

Apart from processing the samples, both short-term and long-term storage conditions are important to ensure both quantity and quality of the samples. The optimal time frame between sample deposition and long-term storage, and the optimal conditions within this time frame, should be defined and standardized within a biobank (Figure 1). For this short-term storage, it is important to determine how long the samples can be at room temperature or 4 °C before long-term storage [23]. For long-term storage, most sample types are known to remain stable at −80 °C [23]. To avoid sample degradation from freezing and thawing, the original sample could be aliquoted in smaller volumes before long-term storage. 

Quality of the methods used to collect and store samples, and thereby the sample quality, should be ascertained by implementing standard operating procedures and quality control systems into the biobank workflow. Although possibly financially unfeasible for small biobank initiatives, large biobanks should consider international ISO9001 and ISO20387 accreditation in order to ensure sample quality [24].

Besides the general considerations that are applicable to all sample types, most non-invasive sample types also have specific properties that require extra awareness. In the following paragraph, we will give some suggestions for general sample collection and storage protocols for liquid biopsies, and elaborate on crucial properties of specific non-invasive sample types which should be taken into account when collecting and processing these samples for storage in a biobank (Figure 1). Although several protocols for liquid biopsy collection and processing have been described previously, the future purpose of the samples should always be taken into account and general protocols should, where applicable, be adapted to that purpose.

### 2.2. Blood

As blood is often collected for routine care purposes, it has proven to be the most popular sample type for non-invasive biomarker studies. For the same reason, protocols for different fractions of blood collection (e.g., whole blood, serum, plasma) and processing have been highly standardized over time (Figure 1) [25,26,27]. For anticoagulated blood, a maximum storage of 24–72 h at room temperature is recommended before long-term storage at low temperatures [28,29,30]. Timing of blood collection does not seem to have an impact on, e.g., DNA or RNA yield; however, in the case of metabolomics analyses, the timing of blood collection should certainly be considered as metabolites are more abundant after high metabolic activity [31].

Whole blood collection requires anticoagulation tubes, of which the most conventional are EDTA and heparin tubes. Despite the fact that heparin tubes are widely used to determine hormone or cholesterol levels in routine clinical care, they are not preferred for molecular analyses. EDTA tubes are preferred over heparin tubes, because of the property to preserve cells and inhibit DNase activity; using these tubes will yield higher DNA concentrations of equal quality [32]. Although high concentrations of EDTA can inhibit PCR efficiency by depleting magnesium ions, heparin is known to act as a PCR inhibitor at much lower concentrations, suggesting that EDTA tubes should be preferred for blood collections aimed at molecular or DNA research (Figure 1) [22]. Next to whole blood, fractions like serum or plasma can be used for molecular analyses. Serum or plasma is preferred over whole blood for cell-free DNA (cfDNA), protein or hormone analyses because the removed cellular (solid) fraction cannot interfere with the results. Although it shares most of its characteristics with respect to PCR inhibitors and DNase/RNase activity with whole blood, serum and plasma are mostly depleted of the known PCR inhibitor hemoglobin [33]. Therefore, serum or plasma collected in EDTA tubes seem to be the preferred sample types for storage in a biobank aimed at molecular biomarkers (Figure 1).

Blood collection requires venipuncture, certified skills and special collection tubes, and is therefore considered the most invasive of all non-invasive samples (Figure 1) [31]. Nonetheless, as blood collection is often part of routine clinical care, collecting an additional blood sample for storage in biobanks does not substantially burden the patient. Therefore, biobanking blood samples is relatively uncomplicated. Examples of large biobanks with blood sample collections from cancer patients are the UK Biobank [34], Biobank Japan [35], the Victorian Cancer Biobank [36], and the Canadian Tumor Repository Network [37].

### 2.3. Feces

Although standardization of blood collection and processing has been established over time, that does not hold true for other non-invasive sample types, like feces. Even though several articles describe collection and processing protocols for feces, no standardized protocol is currently available [38,39,40]. Generally, fecal samples are recommended to be stored at room temperature for a maximum of 24–72 h before long-term storage at low temperatures [41,42,43]. Several molecular components can be analyzed from feces, such as human and microbial DNA, proteins and metabolites; however, collection of whole stool samples requires relatively large containers and thereby poses a logistical challenge. The collected feces samples need to be homogenized, preferably using a buffer stabilizing the samples’ components before freezing. PCR inhibitors to take into consideration when later processing stool samples mainly include bile salts and complex polysaccharides, which could be inactivated by addition of an absorbing buffer (mostly provided in commercial DNA isolation kits), or using a Taq polymerase, which is insensitive to these substances in subsequent PCR experiments (Figure 1) [22]. The addition of a PCR facilitator, like spermidine, could partially overcome PCR inefficiency due to the above-mentioned PCR inhibitors [44]. Other challenges when processing fecal samples are the abundance of, e.g., bacterial DNA over host DNA and undigested debris (Figure 1). Depending on dietary intake and its biological variability, the inorganic fraction of feces contains 25%–54% bacterial biomass, which could interfere with multiple downstream analyses, including human fecal DNA and protein extraction [17,45]. Additional cleanup steps are recommended when processing fecal samples in order to overcome these issues. For DNA isolation, adding an additional precipitation step to purify the sample before starting the extraction using a commercial kit could increase yield. In addition, the technology behind the chosen DNA/RNA/protein isolation kit may influence the yield. Cleanup of these isolations are often based on spin columns; however, an additional DNA/RNA/protein capturing step involving, e.g., magnetic beads could further purify the sample [46]. Notably, bacterial enzymes may contribute to degradation of human DNA and RNA in fecal samples [47].

From a biobank sample-storage perspective, fecal samples tend to take up large amounts of space because of their volume, which is a financial burden as well. An advantage of feces collection is that it can be executed at home, as it does not require specialized skills or (apart from relatively large collection containers) equipment (Figure 1) [31]. Although fecal samples have not been collected from cancer patients in any large publicly funded biobank setting yet, it is becoming a more popular sample type in research settings for several cancer-related studies, as well as for fecal microbiota transplantation studies, as summarized by Terveer et al. [48].

### 2.4. Urine

Although urine has been used to diagnose several diseases and infections for a long time, measuring molecular biomarkers for cancer in urine has only emerged in recent decades. Therefore, similar to feces, no standardized way of collecting and processing urine for storage in biobanks has been established yet. Generally, storing urine at room temperature for a maximum of 4 h is recommended before long-term storing at low temperatures [49,50]. A general protocol for urine collection and storage was published by the UK biobank [26,27]. Molecular components like DNA, RNA, proteins and metabolites can all be measured in urine, but the quantity of these components in urine fluctuates throughout the day. In the morning, a concentrated urine sample can be obtained in terms of, e.g., DNA; in contrast, metabolites are more abundant after high metabolic activity (Figure 1). Although urea can act as a PCR inhibitor, its concentration is usually too low to affect future analyses (Figure 1) [22]. Depending on the purpose of the samples, urine can be centrifuged to separate the cellular fraction and the supernatant, containing tumor-derived cfDNA. After separation, both fractions can be stored separately (Figure 1). Su et al. described that different types of cfDNA could be found in urine; only low molecular weight DNA was derived from the tumor, while high molecular weight DNA was not [51]. It was suggested that only the small-sized DNA fraction in urine, rather than the total DNA, should be used to increase assay sensitivity for cancer-related DNA biomarkers in urine [51].

Storing urine in biobanks has some advantages over other sample types, as urine collection and processing does not require much time, effort or specialized equipment, and home-sampling is a possibility. Moreover, urine is a liquid biopsy that is non-invasive to obtain, which will increase the willingness of participants to donate samples, allowing the establishment of relatively large collections within a limited amount of time (Figure 1). Examples of large biobanks that have collected urine samples from cancer patients are the UK biobank [34] and the Canadian Tumor Repository Network [37].

### 2.5. Other

Less conventional non-invasive sample types include sputum, saliva, mucinous swabs, hairs, nails and exhaled breath. Sputum, saliva and mucinous swabs are liquid samples thought to carry cells that shed from epithelial lining, which is a non-invasive way to examine a potentially cancerous site. Solid sources like nails or hairs have proven to be especially useful for retrospective metabolite or protein evaluation (widely used in forensic sciences) [52,53], but DNA or RNA can also be isolated from these samples [54]. A potential threat of these solid sources is that the hair or nail grows over time, with potentially changing exposures, and depending on the measurement, the results should be interpreted with caution. Moreover, these sources can exclusively be used to examine germline DNA/RNA, rather than cancer-induced changes [55]. Although exhaled breath has been used to study volatile organic compounds as an indicator of cancer, clinical trials with standardized sampling methodology are required before this technique can be implemented for non-invasive cancer biomarkers [56,57,58].

## 3. Using Existing Biobanks for Validation of Potential Molecular Biomarkers

Apart from relatively small research-driven biobanks established by researchers, several very extensive and renowned, often publicly funded, biobanks that are accessible for research purposes exist. Examples of such biobanks have been summarized by Vaught et al. and Patil et al. [59,60], respectively. Although these biobanks carry many valuable samples and data, only a few biobanks include liquid biopsies and are oriented towards cancer research. The UK Biobank [34], BioBank Japan [35], the Victorian Cancer Biobank [36] and the Canadian Tumor Repository Network [37] are examples of large biobanks which include non-invasively collected samples from cancer patients. Existing biobanks could for example support translational research aimed at identifying those patients that would benefit from immune therapy for cancer as they could facilitate relatively fast validation of promising research data obtained through pilot studies.

### 3.1. Level of Evidence

The Level of Evidence (LoE) represents the current evidence for clinical utility of a biomarker, with LoE I representing the highest evidence and LoE V representing the poorest evidence for clinical utility of that particular biomarker [61]. We previously demonstrated that after initial publication of a potential biomarker, subsequent studies do not substantially add to the LoE and clinical translation of potential biomarkers [8]. The quest for novelty has become a paradigm for many researchers; however, this will not benefit the biomarker field in terms of clinical translation. It is, therefore, encouraged to further validate published biomarkers that showed promising results upon initial publication, in order to bridge biomarkers from initial publication to clinical use, and subsequently reduce research waste. To improve the LoE, prospective cohorts including sufficient, appropriate samples are required. Selecting appropriate samples from biobanks that fit the research design could facilitate relatively fast validation (Figure 1). Especially in cases where researchers have easy access to such large cohorts of non-invasive samples, an effort should be made to ensure validation of previously published potential biomarkers independent of the initial research group as this is an essential step in obtaining a sufficient LoE for a potential biomarker [62].

### 3.2. Sample Selection

As biobanks will generally include heterogeneous patient populations (including all TNM stages and grades in case of cancer patients’ samples), it is important to select only those samples from the biobank that fit a specific research question (Figure 1). For early detection of cancer, for example, it is crucial that the biomarker is also assessable in early stage and grade cancers; analyzing subgroups could therefore reveal the true potential of a biomarker. To both select an appropriate sample population from a biobank, and perform subgroup analyses, the availability of correct and adequately annotated clinical data to complement the biological samples is of great importance.

### 3.3. Standardization

Generally, it is considered important to have a certain level of standardization regarding processing and storage of liquid biopsies within and between biobanks to be able to perform validation experiments [4,5]. Although standardized processing and storage within one biobank is a requirement, strict standardization between different biobanks is practically and logistically challenging. However, preferably, biomarkers should be robust and perform equally, independent of the sample pretreatment or assessment procedure, as illustrated by routinely used biomarkers evaluated in various sample types with various techniques, such as *KRAS* and *BRAF* mutations in colorectal cancer [63]. Standardization regarding methods and sample quality can be achieved by implementing standard operating procedures, and by adhering to ISO9001 and ISO20387 accreditation, as described previously [24].

## 4. Biobank Sustainability

Although funding providers might presume that, after initial investments, biobanks should be self-sustainable, this is challenging for multiple reasons. Ensuring solid sources of funding, standardizing procedures to assure sample quality, and complying with legal and privacy-related regulations are critical factors to ensuring biobank sustainability (Figure 1). Establishing biobanks in a way that they adhere to the FAIR (Findable-Accessible-Interoperable-Reusable) principles could encourage biobank sustainability [64].

Underestimating the costs of establishing and maintaining a biobank poses a problem, as these include not only storage costs, but also, e.g., employees, soft- and hardware, maintaining and replacing ultra-cold freezers, and storage room rental [65]. For sustainability, it is crucial to make sure that the biobank is cost-effective and visible, promoting the biobank is necessary to generate additional financial funds [65,66,67].

Standardization of protocols and procedures to ensure quality is advocated in all facets of biobanking. Accreditation and standard operating procedures in a biobank could increase use by researchers, but could also increase the willingness of participants to donate samples [65,66]. Standard operating procedures can also provide adherence to local, national and international law and regulations [65]. Standardization and accreditation could result in the interoperability of samples, meaning that samples from different biobanks could be pooled to increase the statistical power of a study. Although accreditation poses several advantages for biobank sustainability, it might not be feasible to implement for small biobanks because of accreditation costs.

To stimulate funding acquisition, well thought-through cost-benefit analyses, preliminary data obtained from the biobank and close partnership with biobank users, who could include funding requests for biobanking in their grant proposals, are desirable [65].

## 5. Conclusions

This perspective describes some of the challenges in biobanking, both in general and dependent on the collected sample type, also summarized in Figure 1. Biobanks could facilitate relatively fast validation of research findings like diagnostic biomarkers for cancer, provided that the used samples match the research question. Considering the future purpose of samples is crucial before implementing standardized procedures and logistics to process and store them. Interoperability of samples from different biobanks could facilitate larger sample sizes and thereby increase statistical power of a study. However, as biomarkers should be robust, the degree of standardization between biobanks necessary for biomarker research remains uncertain. Next to establishing biobanks, researchers should make an effort to use existing biobanks and ensure independent validation of previously published potential biomarkers, as this is an essential step for clinical implementation of biomarkers. Apart from technical and methodological considerations, biobank sustainability should be considered throughout all phases of biobanking. We therefore highly recommend that biobanks adhere to the FAIR principles and register in directories like https://directory.bbmri-eric.eu/for European biobanks and https://specimencentral.com/biobank-directory/ for large biobanks worldwide to promote visibility and stimulate use.

## Figures and Tables

**Figure 1 cancers-12-00776-f001:**
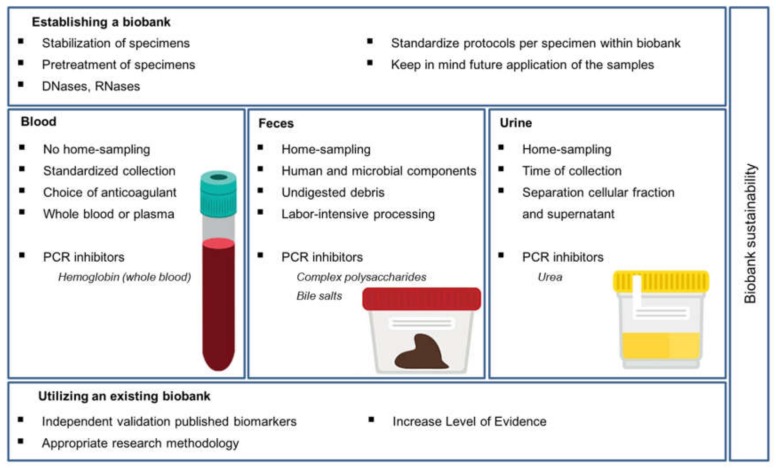
Summary of crucial considerations for establishing a biobank, handling of specific sample types and using existing biobanks.

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
