# Peer review of "Biobanking in Molecular Biomarker Research for the Early Detection of Cancer"

_cancers, 2020, doi:10.3390/cancers12040776_

Round 1

Reviewer 1 Report

Given the importance of biobanking, this paper is a rare and timely paper.

  1. Let's add a difference in terms of utility between general biobanking and biobanking specialized in specific diseases.
  2. Could you recommend general collection protocols in liquid biopsies in examples ?
  3. It is needed the principle of quality control  and real method.
  4. What is the maximum amount of time you can expose to room temperature?
  5. How about adding biobanking services to help researchers in other countries?

Author Response

Thank you for reviewing our manuscript and the comments provided. We have taken these into account when revising the manuscript.

Reviewer 2 Report

Biobanking in molecular biomarker research for the early detection of cancer is an interesting manuscript for those interested in validating biomarkers. Nevertheless, examples regarding usefulness in the cancer area would enrich the paper. 
More illulstrative diagrams with clinical implications in sample collection and processing would improve the quality of the manuscript.

A brief description of translational medicine application of biobanks for molecular biomarkers  would attrack more readers 

Author Response

(The authors gave the same response as above.)
